# Estimation of Body Weight Based on Biometric Measurements by Using Random Forest Regression, Support Vector Regression and CART Algorithms

**DOI:** 10.3390/ani13050798

**Published:** 2023-02-22

**Authors:** Cem Tırınk, Dariusz Piwczyński, Magdalena Kolenda, Hasan Önder

**Affiliations:** 1Department of Animal Science, Faculty of Agriculture, Igdir University, 76000 Iğdır, Türkiye; 2Department of Animal Biotechnology and Genetics, Faculty of Animal Breeding and Biology, Bydgoszcz University of Science and Technology, 85-796 Bydgoszcz, Poland; 3Department of Animal Science, Ondokuz Mayis University, 55139 Samsun, Türkiye

**Keywords:** random forest, support vector machine, CART, biometric measurements

## Abstract

**Simple Summary:**

This study aimed to estimate body weight from various biometric measurements and features such as genotype (share of Suffolk and Polish Merino genotypes), birth weight (BiW), sex, birth type and body weight at 12 months of age (LBW) and some body measurements such as withers height (WH), sacrum height (SH), chest depth (CD), chest width (CW), chest circumference (CC), shoulder width (SW) and rump width (RW). Three hundred and forty-four animals were used in the study. Data mining and machine learning algorithms such as Random Forest Regression, Support Vector Regression and classification and regression tree were used to estimate the body weight from various features. Results show that the random forest procedure may help breeders improve characteristics of great importance. In this way, the breeders can get an elite population and determine which features are essential for estimating the body weight of the herd in Poland.

**Abstract:**

The study’s main goal was to compare several data mining and machine learning algorithms to estimate body weight based on body measurements at a different share of Polish Merino in the genotype of crossbreds (share of Suffolk and Polish Merino genotypes). The study estimated the capabilities of CART, support vector regression and random forest regression algorithms. To compare the estimation performances of the evaluated algorithms and determine the best model for estimating body weight, various body measurements and sex and birth type characteristics were assessed. Data from 344 sheep were used to estimate the body weights. The root means square error, standard deviation ratio, Pearson’s correlation coefficient, mean absolute percentage error, coefficient of determination and Akaike’s information criterion were used to assess the algorithms. A random forest regression algorithm may help breeders obtain a unique Polish Merino Suffolk cross population that would increase meat production.

## 1. Introduction

Sheep play an important role in both obtaining animal products and developing the rural economy among civilizations [1,2]. In addition, sheep need a shorter time between generations than cattle. As in all farm animals, environmental factors play an essential role in the interaction and genetic potential of the sheep. Genotype, the environment and their interaction are the factors that must be considered to make an economical profit. To achieve a high-level yield, it may be necessary to use different genotypes or crossbreeds, taking into account environmental factors. In 2020, 31 breeds and lines of sheep were used in Poland, and the largest share in the breed structure was Polish Merino sheep [3].

The origin of merino sheep breeding in Poland dates back to the 19th century. Merino sheep gradually evolved in terms of thicker wool and improved meat characteristics. Many years of breeding have resulted in breeds that have been genetically combined and used for meat and milk production. Polish Merino has a close and dense fleece. Additionally, Polish Merino sheep mature early and show an aseasonality in reproduction. Because of these characteristics, the Polish Merino is Poland’s most common commercial breed. High-quality meat characteristics describe the Polish Merino sheep, and lambs can be used for dairy and medium-intensity and intensive fattening processes. The body weight of Polish Merino ewes and rams is 55–75 kg and 90–120 kg, respectively [4]. The average fertility of ewes was 94.04% according to Piwczyński [5], and fertility was 152% according to Piwczyński and Mroczkowski [6].

In the 1990s, the Polish Merino sheep breed was improved by crossbreeding with other breeds to enhance some characteristics; therefore, the number of native purebred sheep decreased considerably. Consequently, in 2008 the pure Polish Merino sheep breed was characterized, and the original breed pattern (maintains the breed purity) was described. From then, the breed was called Polish Merino sheep [7]. The crossbreeds of Polish Merino ewes that appeared in the breeding stock books in 2015–2020 decreased from 3.93 to 1.71%, while the pure Polish Merino ewes were relatively stable (2015: 10.65%; and 2020: 10.69%) [3].

The only condition for profitability in sheep farms in Poland is producing young lamb for slaughter. Unfortunately, Poland’s sheep population structure does not meet the requirements for producing meat lambs. Sheep used for their wool are the most numerous, while the stock of meat breeds is scarce: in 2020, the number of Suffolk ewes under the utility assessment was only 151 [3]. The use of displacement crossing is a breeding method that can change the breed structure in favor of meat breeds [5]. The backcrossing of dams of Polish Merino sheep with meat breeds rams, among others Suffolk, might be an efficient way to increase the meat sheep population in Poland [8].

The Suffolk breed originated in England and was created by crossing the Southdown rams with the Norfolk Horn ewes. The breed was recognized in 1810. In 1886, the Suffolk Breeders’ Association started to keep a register of animals of this breed. Thanks to the intensive selection and proper selection of breeding pairs, animals with outstanding meat characteristics were produced. They were suited for crossing with other breeds to improve fattening and slaughter characteristics. The literature shows many examples of crossbreeding Polish Merino and Suffolk sheep [8].

Growth and development are economically important features. Growth is determined by measurement and weighting, and its calculation is based on live weight. Furthermore, growth and weight gain can be calculated using various correlations between live weights and body measurements [9]. Body weight is the most important feature for all animal species with an economic income, as it directly affects breeding income and meat production. Scientifically, more interest has been paid to defining the association between body measurements and body weight in improving meat production. Biometric measurements of all animals may indicate phenotypic and genotypic characteristics as well as growth characteristics [10]. It has been reported that various biometric features taken throughout the early growth periods may be used as an early selection criterion to obtain offspring with superior body weight [11]. In addition, body measurements are helpful in herd management in estimating body weight. Furthermore, being aware of body weight is essential in herd management practices such as calculating the amount of feed per animal, managing the medicinal drug doses and determining the optimum slaughter weight [12,13]. For this reason, it may be used as an indirect selection criterion in making predictions based on body measurements [1,14]. In this framework, the finest way to determine biometric measurements that directly affect body weight and define breed phenotypically is the application of trustworthy statistical procedures, such as multivariate statistical procedures for sheep [15,16]. There may be differences in estimating live weight from body size from species to species and from breed to breed. Many studies evaluate body weight using measurements in several animal species, such as buffalo, sheep, dogs, cattle, goats, rabbits and camels [14,17,18,19,20,21,22,23,24,25,26,27].

The three methods selected are the subject of multivariate statistics. Regression analysis is one of the multivariate statistical methods used to reveal the relationship between biometric features and animal weight. In multivariate statistical modelling, regression analysis is a process to estimate the relationship between explanatory and response variables. Many methods are used to estimate the response variable, with the most common being the Least Squares (LS) method. The LS method requires some assumptions to make an effective model estimation. Alternative methods are proposed when multicollinearity between explanatory variables is provided from these assumptions [28,29]. Many studies in different species and breeds are based on estimating body weight using biometric features. They use multiple regression, Classification and Regression Tree (CART), Chi-square Automatic Interaction Detector (CHAID), Multivariate Adaptive Regression Splines (MARS) algorithms and artificial neural networks. Although there are different studies for other breeds and species, there is no such study for different shares of Polish Merino and Suffolk genotypes in crossbreeds, which is the subject of our study [1,12,16,30,31,32,33]. In addition, there is no study for CART, SVR and RFR, apart from the scope of the aforementioned algorithms. In this framework, various statistical procedures can help produce more reliable estimates covered by indirect selection criteria in different animal species and expose the biometric features that influence body weight. The use of many methods such as CHAID, MARS, CART, Artificial Neural Networks (ANNs) and Exhaustive Chi-square Automatic Interaction Detector (Exhaustive CHAID) has gained importance in estimating body weight from biometric features in various sheep breeds [1,16,24,32,33,34]. The use of these estimation methods, however, differs from breed to breed. To our knowledge, there is no research on using CART, Support Vector Regression and Random Forest Regression algorithms for body weight estimation of the different shares of Polish Merino in the genotype of crossbreeds. These three methods were selected to show a clear presentation of the results. The present study aims to fill this gap in the literature and evaluate the goodness of fit of these procedures.

## 2. Materials and Methods

The numerical material used in the research came from the research carried out in 1990–1995 by Piwczyński [35] as part of the topic of his master’s and doctoral dissertation. The research material consisted of 344 animals, including 114 crossbreds R2 (75.0% Suffolk, 25.0% Polish Merino), 97 crossbreds R3 (87.5% Suffolk, 12.5% Polish Merino) and 133 animals of Suffolk breed. A total of 88 rams and 256 sheep were used in the study. The evaluated groups of crossbreds originated from the two subsequent stages of backcrossing of Polish Merino ewes with Suffolk rams. Implementation of crossbreeding started in 1986 in one flock in Bydgoszcz voivodship. The purpose of this crossbreeding was to obtain a meat-type sheep line.

Suffolk sheep used for crossbreeding with Polish Merino were imported to the farm from Great Britain in 1985. The group of animals consisted of 40 ewes and five rams. All test animals were housed in litter-box buildings with running water and artificial lighting. Mothers and lambs were fed in accordance with the applicable nutrition standards declared by the National Research Institute of Animal Production, 1985. During the summer feeding period, the animals used a pasture. While on-site, they were fed a CJ mixture (for calves and lambs), dried corn, hay and green alfalfa, and during the winter feeding they were given a CJ mixture, beets, oats, dry pulp, briquette haylage and hay.

To compose the data set, the genotype (share of Suffolk and Polish Merino genotypes), birth weight (BiW), sex, birth type and body weight at 12 months of age (LBW) and some body measurements such as withers height (WH), sacrum height (SH), chest depth (CD), chest width (CW), chest circumference (CC), shoulder width (SW) and rump width (RW) were used [4].

Algorithms such as CART, CHAID and Exhaustive CHAID are tree-based algorithms used to evaluate a quantitative feature [14,16,36,37]. For this purpose, Breiman et al. [38] developed the first method called CART procedure. The CART algorithm is a binary decision tree structure created by recursively dividing a node into two child nodes. The algorithm covers the evolving process until many homogeneous nodes are obtained from a learning sample dataset, providing minimal error variance covered by training and test sets.

The main purpose of a tree is to select new and homogeneous binary splits to obtain the purest child nodes. In the algorithm, each split is made for one estimator only. The variance-based method was used as the pruning rule in the tree construction, and the minimum size of a tree node was set to five and accepted as the stopping criterion. In addition, 10-fold cross-validation with a single standard error rule was applied to find the regression tree that fit the training data. In this way, it was warranted that there was no overfitting for the CART algorithm.

A valuable part of the support vector machine (SVM), one of the most commonly used procedures among machine learning procedures, is the support vector regression (SVR) procedure [39]. In the SVM algorithm, the part that deals with classification is known as support vector classification (SVC), and the part that deals with prediction is known as SVR [40,41,42]. The SVR is one of the machine learning procedures that creates a linear model prediction to simultaneously minimize experimental risk and model complexity [43]. While SVR is a regulated learning procedure, the presentation of SVR varies according to the training and test sets [44].

The primary theory of SVR is to describe a function f(x) with the upper limit deviation (ε) from the train set. The training set points are arranged inside the cutoff point between −ε to +ε [44]. However, most studies cannot be modeled in a linear sense. Therefore, for conditions for which the solution is nonlinear, the input data of the SVR algorithm is mapped to a better dimensional Hilbert space (H), so the edge of the regression model can be linear [39].

The regression hyperplane to be achieved under nonlinear conditions is presented below.
$$\hat{y}=\langle w,\phi \left(x\right)\rangle +b$$
where, w is the weights of the vector, ϕ(x) is the functions of the kernel, 〈.,.〉 implies an innermost vector result and b is a biased term. In addition, many kernel functions can be used to apply to nonlinear conditions. Although there are many kernel functions, the Gaussian radial basis function is used in this study.

Random Forest is a standard procedure used between several multivariate statistical procedures in terms of its practicality for regression and classification form of the problems. The RFR algorithm consists of a process that includes a layer of casualness to the bagging algorithm. This procedure was presented by Breiman [45]. The RFR algorithm is shown as a set of limitations utilized hierarchically from the tree’s root to the leaf by merging clusters of the regression trees [46,47]. The most significant benefit of this procedure is that it can be clearly utilized in nonlinearity.

The procedure requires a method that consists of three stages [48]. The first stage is constructing a number of trees (n_tree_) from the initial data. The second stage is to build an untrimmed regression or classification tree for every sample. The final step is predicting the recent data of the tree. In this aspect, for the Polish Merino sheep and Suffolk crossbreed sheep data set, the model parameters such as n_tree_ and number of variables checked out for all splits are chosen (m_try_) as 500 and 5, respectively.

To compare the model performances, the goodness of fit of criteria such as the Pearson correlation coefficient (r), root mean square error (RMSE), determination of coefficient (R^2^), Akaike’s information criterion (AIC), mean absolute percentage error (MAPE) and standard deviation ratio (SD_ratio_) were used as shown below [49,50]:Root-mean-square error (RMSE):
$$\mathrm{RMSE}=\sqrt{\frac{1}{n}\overset{n}{\underset{i=1}{\sum }}{\left(y_{i}-y_{\mathrm{ip}}\right)}^{2}}$$Akaike information criterion (AIC):
$$\{\begin{array}{c} AIC=n.ln\left[\frac{1}{n}\overset{n}{\underset{i=1}{\sum }}{\left(y_{i}-y_{ip}\right)}^{2}\right]+2k,\;\mathrm{if}\;n/k>40\;\\ \;AIC_{c}=AIC+\frac{2k\left(k+1\right)}{n-k-1}\;otherwise \end{array}$$Standard deviation ratio (SDR):
$${\mathrm{SD}}_{\mathrm{ratio}}=\frac{S_{m}}{S_{d}}$$Global relative approximation error (RAE):
$$\mathrm{RAE}=\sqrt{\frac{\sum_{i=1}^{n}{\left(y_{i}-y_{\mathrm{ip}}\right)}^{2}}{\sum_{i=1}^{n}y_{i}{}^{2}}}$$Mean absolute percentage error (MAPE):
$$\mathrm{MAPE}=\frac{1}{n}\;\overset{n}{\underset{i=1}{\sum }}\left|\frac{y_{i}-y_{\mathrm{ip}}}{y_{i}}\right|*100$$
where, n is the number of the training data, y_i_ is the actual amount of the response variable (BW), $y_{\mathrm{ip}}$ is the estimated amount for the response variable (BW), S_d_ is the standard deviation for the response variable (BW) and S_m_ is the standard deviation of the best model’s errors. The aforementioned goodness of fit was used to compare the model performances, which were made along with the lowest RMSE, SD_ratio_ and MAPE. In addition, the model performances evaluated the highest r and R^2^ values [51].

## 3. Results

In this study, the mean and standard deviation for each trait for Polish Merino sheep and Suffolk crossbreed sheep were calculated, and the descriptive statistics are presented in Table 1.

The correlation coefficient was used to define the association between body measurements (birth weight, withers height, chest depth, sacrum height, chest width, shoulder width, rump width and chest circumference) and sex, birth type characteristics and LBW. Figure 1 shows the most significant correlation coefficient between CC and LBW (coefficient of 0.72). The other traits, except for the SH, show a similar correlation coefficient with LBW. Moreover, all coefficients were determined to be significant (*p* < 0.05).

The tree diagram constructed using the CART algorithm is presented in Figure 2. The root node of the tree was recorded as 56 kg (LBW). In the case of CC, the initial depth was lower than 94 cm, and the average LBW of the crossbreed sheep was determined to be 49 kg. At the right side of the tree, the initial depth, in the case of CC, was greater than 94 cm, and the average LBW was determined as 63 kg. If the CC was less than 94 cm in the first split, the tree was divided into 2 parts. The first part was for the sex. If the sex was female, the tree was divided for CC < 89 and CC ≥ 89. In these cases, LBW was determined as 46 and 51 kg, respectively. If CC was ≥ 94 cm in the initial split, the tree was divided into 2 parts: CD < 32 cm and CD ≥ 32 cm. If CD was <32 cm, the tree was divided for Genotype = R2, WH < 63 cm, SW < 29 cm and RW < 26 cm. The average LBW was determined as 66 kg for the cases in which the Genotype = R2, WH < 63 cm, SW < 29 cm and RW < 26. In the case of the CD ≥ 32 cm, the average LBW was determined as 73 kg. This node was divided into two parts for Genotype = R2. If the genotype was R2, the average LBW was defined as 62 kg. However, when the genotype was not R2 (i.e., was R3 or Suffolk) and CD ≥ 34 cm, the LBW was determined as 88 kg (the node with the highest values of LBW). To contribute to the rural economy by increasing meat productivity, it has been determined that more profitable livestock can be made in cases when CC ≥ 94 cm, CD ≥ 32 cm, the genotype is not R2 and CD ≥ 34 cm.

First, the SVR procedure was performed for the training set. After the training procedure, the SVR predicted the body weight of Suffolk sheep. The kernel function was estimated for the final body weight. The accessibility for the model is based on the selected factors such as epsilon and cost (C). The aforementioned factors were examined for several values, and the procedure was utilized for the epsilon and C values, which would provide a highly trustworthy model. Sensitivity analysis was used to estimate the model’s virtual significance for explanatory variables for BW (Figure 3). CC had the most significant relative importance value obtained in the scope of the sensitivity analysis. The explanatory variable that produced the smallest relative importance value was genotype and birth type (twin).

The RFR algorithm performance is presented in Table 2. Moreover, the sensitivity analysis was performed to predict virtual significance amounts of the explanatory variables LBW in RFR (Figure 4). For sensitivity analysis, CC, CD and SW had virtual significance higher than 10%. However, unlike the SVR algorithm, the explanatory variables that produced low virtual significance were determined to be CW and SH. The lowest virtual significance had BiW.

The comparison of all algorithms and goodness of fit criteria are presented in Table 2. For all algorithms, the performances of the training and the test sets were evaluated. The performance values obtained from the test set for each model were weaker than those from the training data set. The best procedure for the test set was the RFR algorithm, although the most appropriate algorithm for the training set was SVR.

RFR was determined as the most appropriate algorithm as it gave closer results for the training and test sets and gave the highest R^2^ and r values and the lowest RMSE, SD_ratio_, CV, MAPE and AIC values in the test set. Because the training set memorized the SVR algorithm, the SVR algorithm gave unreliable test results.

## 4. Discussion

Different characterization methods are used in the literature to investigate the relationship between biometric features and body weight in various animal species [52]. The accuracy of statistical methods applied to predict BiW from biometric features that differ even between breeds is also very important. Many studies have been conducted on this subject in different animal species and breeds. This subject is very important, especially in rural areas and conditions where no weighing device is available [52]. However, there is no study on this aim for the Suffolk sheep breed. In multivariate statistics, artificial neural networks, data mining, machine learning algorithms and the usage of the goodness of fit criteria have been suggested in selecting the finest model [18]. Within this scope, the model performances are compared according to the goodness of fit criteria [51].

CART, SVR and RFR algorithms were used to help determine the selection scheme for Polish Merino sheep and Suffolk crossbreed sheep. Various statistical methods can define effective variables for LBW estimation, which may be helpful for selecting farm animals; therefore, the basis for the sustainable animal breeding may be laid. Though the literature lacks studies on these algorithms, it has been established that similar to our study, only the RFR and SVR algorithms were used for the Thalli sheep breed [52]. In that study, Tırınk [52] indicated that the MARS algorithm was superior to Bayesian Regularized Neural Network (BRNN), SVR and RFR. The results of this study showed that SVR was better than RFR. That study had the opposite results to ours. It means that the model selection depends on the genotype.

Alonso et al. [42] used Support Vector Machine Regression to estimate the carcass weight in Asturiana de los Valles beef. For this aim, 390 measurements for 144 animals were made. According to these results, the optimal carcass weight prediction was obtained 150 days before the slaughter time. They presented the use of SVR algorithm detailed.

Ali et al. [16] compared the CART, CHAID, ANNs and Exhaustive CHAID algorithms in this study for the Harnai sheep breed. The results were estimated as follows: Exhaustive CHAID 0.8421, CHAID 0.8377, ANNs 0.81999 and CART 0.82644. When the performance of the CART algorithm was evaluated against other algorithms within the scope of R^2^, the diversity of the algorithms used and the breed differences, the CART algorithm was the third-best algorithm. However, it gave results close to other algorithms in terms of performance. Our result showed that both SVR and RFR were better than CART algorithm. According to this, SVR and RFR algorithms had better fitting properties than CHAID and ANNs.

Celik et al. [1] aimed to compare CART, MLP, CHAID, MARS, Exhaustive CHAID, and RBF for Mengali rams. In the scope of the goodness of fit criteria such as R^2^, RMSE and SD_ratio_, the finest estimation model was defined as the CART algorithm. The only comparable algorithm was CART, which we also used in our study. According to this, the SVR and RFR algorithms may be more appropriate, but it should be considered that the fitting performance may depend on the data.

Hussain et al. [53] compared the hybrid machine learning algorithms such as SVR and emotional ANNs for estimating the body fat percentage. They used anthropometric characteristics (BFP, sex, age, weight, height, WHR, abdominal C and BMI). RMSE, R^2^, and rRMSE were used for the model comparison criteria. According to the BFP results, SVR was 0.9682 for R^2^, 0.0245 for RMSE, and 7.6956 for rRMSE. However, the hybrid (SVR-EANN) method was the best algorithm for estimating the BFP. In our work, SVR was not the best one. The superiority of the RFR algorithm versus the SVR algorithm should be taken into account because the SVR algorithm gave unreliable test results due to it being memorized in the training set.

Iqbal et al. [54] aimed to compare the model performances for gradient boosting machine, regression tree, random forests and SVM algorithms. Beetal goats were used, and the explanatory variables such as sex, body length, shank circumference, neck length, head girth, rump height and belly sprung were evaluated. In this study, the model comparison criteria such as Pearson’s correlation, R^2^, MAE, MAPE and RMSE were chosen. According to the results, the gradient boosting machine (GBM) was determined to be the best model for predicting the body weight of Beetal goats. However, the random forest regression algorithm was the second-best algorithm. RFR was one of the best algorithms, as indicated in this study’s results. In our stuy, there is no evidence to compare GBM, which was the best for the Iqbal et al. [54] study, but it is clear that RFR can be used reliably for model fitting.

Marco et al. [55] aimed to examine the AdaBoost ensemble learning method and RFR for different data sets. Several machine learning techniques, such as CART, kNN, MLP, SVR and RFR were used for different data sets. According to the results of most of the datasets applied in the study, they stated that RFR was the most reliable and successful algorithm for their study. The results matched to results of our study.

Ahmad et al. [56] aimed to compare various algorithms such as RFR, decision trees, extra trees and SVR. They wanted to predict solar thermal energy systems and revealed that the RFR and extra trees models gave more reliable results than variable selection tools. The matching Ahmad et al. [56] and Marco et al. [55] results with our study showed that RFR can be used instead of other algorithms.

Coşkun et al. [34] aimed to compare the eXtreme Gradient Boosting (XGBoost), Random Forest (RFR) and Bayesian Regularization Neural Network (BRNN) data mining algorithms to predict the live weight at the end of fattening by using some of the body characteristics at the initiation of fattening in Anatolian Merino lambs. They indicated that the XGBoost algorithm gave a better fitting performance than BRNN and RFR according to the root mean square error (RMSE), standard deviation ratio (SDR), mean absolute percentage error (MAPE) and adjusted coefficient of determination (R^2^_Adj_). For the interpretation

Coşkun et al. [34] focused on short-term fattening performance results. Even though not the best algorithm, the CART algorithm showed that the crossbreeding level should be at least R3 to reach the highest live body weight in Polish conditions.

Compared to previous research results, several species and breeds were used for data mining and machine learning algorithms. The animal age, differences in flock management systems and differences in statistical methods applied can be attributed to the extensive variation in previous studies. When comparing our study with other results, the models we used according to the selected goodness-of-fit criteria give similar results as other studies. However, recommending several statistical procedures for BW estimation using biometric features is important in terms of both species and breed characterization for meat production industries. It reveals that more studies are needed on this subject.

## 5. Conclusions

To conclude, the RFR procedure may help breeders improve the characteristics of great importance. Moreover, it shows BW as a criterion for establishing proper biometric measurements and flock organization principles. The study’s outcomes showed that based on the goodness of fit criteria for choosing the most appropriate model, machine learning and data mining algorithms can be profitably utilized for body weight prediction based on measured body measurements.

## Figures and Tables

**Figure 1 animals-13-00798-f001:**
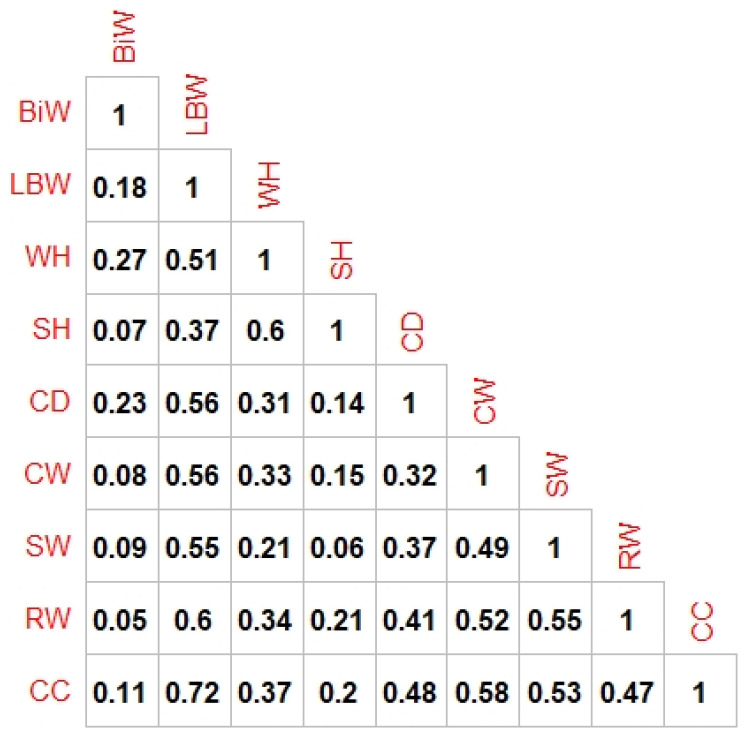
Correlation matrix.

**Figure 2 animals-13-00798-f002:**
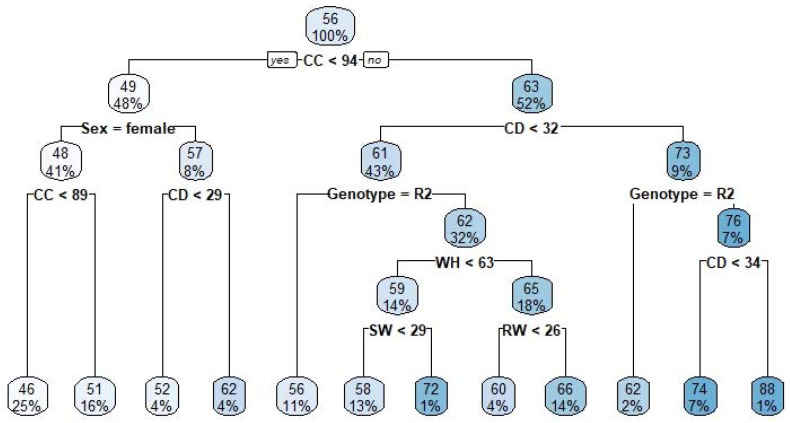
The constructed CART diagram.

**Figure 3 animals-13-00798-f003:**
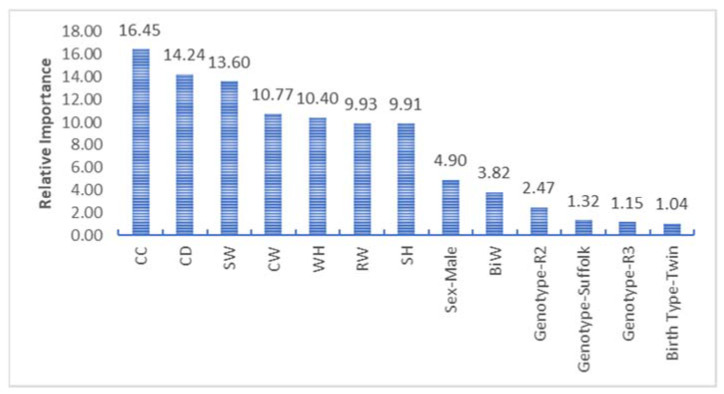
Relative importance for the SVR algorithm.

**Figure 4 animals-13-00798-f004:**
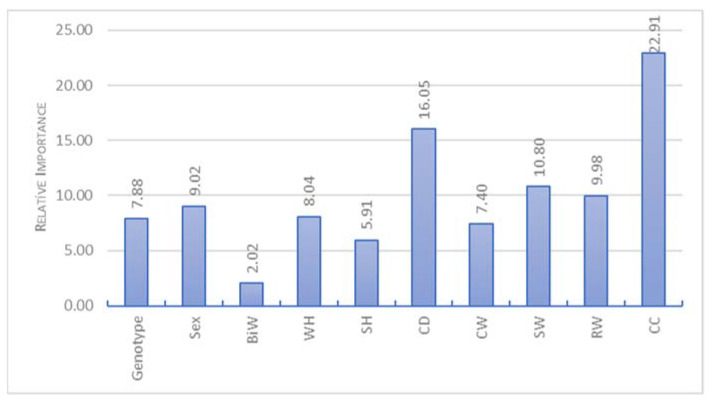
Sensitivity analysis for RFR algorithm.

**Table 1 animals-13-00798-t001:** Descriptive statistics.

Genotype	Variables	Mean ± Standard Deviation
Suffolk N = 133	BiW	3.79 ± 0.92
	LBW	58.32 ± 10.32
	WH	62.07 ± 3.30
	SH	64.49 ± 4.08
	CD	28.86 ± 2.47
	CW	22.95 ± 2.65
	SW	24.65 ± 2.68
	RR	26.54 ± 2.96
	CC	94.16 ± 8.29
R2 N = 114	BiW	4.22 ± 0.82
	LBW	52.70 ± 6.74
	WH	63.18 ± 2.58
	SH	63.61 ± 2.45
	CD	28.50 ± 4.39
	CW	23.12 ± 2.60
	SW	22.71 ± 4.39
	RR	25.20 ± 2.77
	CC	92.64 ± 6.90
R3 N = 97	BiW	4.13 ± 0.96
	LBW	58.77 ± 11.56
	WH	62.66 ± 3.34
	SH	63.35 ± 3.39
	CD	29.52 ± 2.26
	CW	23.20 ± 2.50
	SW	24.90 ± 2.59
	RR	26.35 ± 2.93
	CC	95.93 ± 9.30

Birth weight (BiW), sex, birth type and 12th month of body weight (LBW) and some body measurements (cm) such as withers height (WH), sacrum height (SH), chest depth (CD), chest width (CW), chest circumference (CC), shoulder width (SW) and rump width (RW).

**Table 2 animals-13-00798-t002:** The results of the CART, SVR and RFR algorithms in the scope of the goodness of fit criteria.

Criterion	CART		SVR		RFR	
	Training Set	Test Set	Training Set	Test Set	Training Set	Test Set
RMSE	20.304	49.750	16.100	33.745	24.271	31.279
SDratio	0.454	0.643	0.404	0.526	0.497	0.511
CV	8.000	12.320	7.110	10.080	8.740	9.780
r	0.891	0.774	0.918	0.852	0.869	0.860
MAPE	6.643	9.801	4.984	6.822	6.848	7.046
R^2^	0.793	0.578	0.836	0.714	0.753	0.735
AIC	830.985	265.677	766.951	239.280	880.245	234.121

## Data Availability

To reach the data please contact with the authors C.T.

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
