# Peer review of "Estimation of Body Weight Based on Biometric Measurements by Using Random Forest Regression, Support Vector Regression and CART Algorithms"

_animals, 2023, doi:10.3390/ani13050798_

Round 1

Reviewer 1 Report

Understanding changes in body weight in animals is essential for feeding and herd management. records body wt are in short supply and body measurements and other body variables help in prediction body weight. Although the authors identified an important problem to study

1. why didn't they try first simple linear and multivariate regressions for prediction and comparison

2. There are several machine learning and data mining algorithms and no explanation has been given why they only focused on the three algorithms

3. If possible, Please present the relative importance of the predictor variables for the Random forest algorithm

4. Please check the English in the manuscript and try to avoid ambigous and long sentences

Author Response

Reviewer Comment

Why didn't they try first simple linear and multivariate regressions for prediction and comparison?

Answer

We didn’t try it because we well know that for this related observations, the multicollinearity problem make the model weak.

The sentences “Studies using many traits are the subject of multivariate statistics. Regression analysis is one of the multivariate statistical methods used to reveal the relationship between biometric features and animal weight. In multivariate statistical modelling, regression analysis is a process to estimate the relationship between explanatory and response variables. Many methods are used to estimate the response variable, the most common being the Least Squares (LS) method. LS method requires some assumptions to make an effective model estimation. Alternative methods are proposed when multicollinearity between explanatory variables is provided from these assumptions [28,29].” were added.

Lines: 124-135

Reviewer Comment

There are several machine learning and data mining algorithms and no explanation has been given why they only focused on the three algorithms?

Answer

In original draft lines between 109 and 114 this information was given.

Also the sentence “This three methods were selected to able to show clear presentation of the results” was added.

Line: 152-153

Reviewer Comment

If possible, please present the relative importance of the predictor variables for the Random forest algorithm.

Answer

It was given in Figure 4 in original draft.

Reviewer Comment

Please check the English in the manuscript and try to avoid ambigous and long sentences.

Answer

Checked and corrected throughout the text.

Reviewer 2 Report

The authors tackle an interesting and economically important subject, which could provide solutions to farmers to improve their income in a country where meat sheep are not very widespread. nevertheless, the available data could be much better used, the results described in a way conducive to the conclusion the authors consider.

The use of statistically innovative methods could be much better explained, leading, again to the conclusion. The discussion is a line of the results/conclusions from the literature and has very little merit in comparing personal results with the cited data.

The article needs serious improvement of the English language (see the attached document), some of the sentences being very difficult to understand and misleading; in some places such sentences diminish the scientific value of the research. 

Author Response

Reviewer Comment

Lines 39-41 “In addition, sheep need a shorter time between generations than cattle. In addition to the genetic potential of sheep, environmental factors also play a significant role in the emergence of quantitative phenotypes in animal breeding.” – could you, please , rephrase ?

Answer

Re-written as;

“In addition, sheep need a shorter time between generations than cattle. As in all farm an-imals, environmental factors have an essential role, interaction, and genetic potential for sheep. Genotype, environment and their interaction are the factors that must be consid-ered in order to make an economical production.”

Lines: 45-49

Reviewer Comment

Lines 42-43 “Considering the environmental factors, it is possible to breed different breeds among the species.” – difficulty to understand what the authors mean by this, please make it more explicit.

Answer

Re-written as;

“Genotype, environment and their interaction are the factors that must be considered in order to make an economical production. To achieve a high-level yield, it may be neces-sary to use different genotypes or crossbreeds, taking into account environmental factors.”

Lines: 51-54

Reviewer Comment

Lines 50 This makes the breeds most commercially useful in Polish Merino dams.

Answer

Re-written as;

“Because of these characteristics, the Polish Merino is Poland's most common commercial breed.”

Lines: 61-62

Reviewer Comment

Lines 53-54 “According to PiwczyÅ„ski [4], PiwczyÅ„ski and Mroczkowski [5], the average fertility of ewes was 94.04%, and fertility 152%.” – please rephrase

Answer

Re-written as;

“The average fertility of ewes was 94.04%, according to PiwczyÅ„ski [5], and fertility was 152%, according to PiwczyÅ„ski and Mroczkowski [6].”

Lines: 65-67

Reviewer Comment

Line 56 What do the authors mean by “productive breeds”? Is Polish Merino non-productive?

Answer

Re-written as;

“The Polish Merino sheep breed was improved by crossbreeding with other breeds to en-hance some characteristics.”

Lines: 69-70

Reviewer Comment

Line 57 “In the 1990s, The Polish Merino sheep was improved by crossbreeding with productive breeds; therefore, the number of pure-breed sheep decreased considerably. Consequently, in 2008 the pure Polish Merino sheep breed was recognised to characterise the original breed pattern (maintains the breed purity) [6]” Difficult to understand what the authors meant

Answer

Re-written as;

“In the 1990s, The Polish Merino sheep breed was improved by crossbreeding with other breeds to enhance some characteristics; therefore, the number of native pure-breed sheep decreased considerably. Consequently, in 2008 the pure Polish Merino sheep breed was characterised, and the original breed pattern (maintains the breed purity) was de-scribed. From then, the breed was called Polish Merino sheep [7].”

Lines: 69-75

Reviewer Comment

Lines 59- 60 A controversial sentence “The share of Polish Merino ewes that appeared in the breeding stock books in 2015-2020 decreased from 3.93 to 1.71%, while the pure Polish Merino ewes were relatively stable: 2015 - 10.65%, 2020 - 10.69%.” which are the Polish Merino ewes in the first part, if not the pure breed?

Answer

Re-written as;

The cross-breeds of Polish Merino ewes that appeared in the breeding stock books in 2015-2020 decreased from 3.93 to 1.71%, while the pure Polish Merino ewes were relatively stable: 2015 - 10.65%, 2020 - 10.69%.

Lines: 75-77

Reviewer Comment

Lines 68-69 “The backcrossing of dams of Polish Merino sheep with meat breeds rams, among others Suffolk, might be an efficient way to increase the sheep population in Poland [7].” – the meet sheep population or the entire population?

Answer

Re-written as;

“The backcrossing of dams of Polish Merino sheep with meat breeds rams, among others Suffolk, might be an efficient way to increase the meet sheep population in Poland [7].”

Line: 86

Reviewer Comment

Line 130 Could you explicit “applicable nutrition standards”?

Answer

Re-written as;

“Mothers and lambs were fed in accordance with the applicable nutrition standards declared by the National Research Institute of Animal Production, 1985. During the summer feeding period, the animals used a pasture. While on-site, they were fed a CJ mixture (for calves and lambs), dried corn, hay, and green alfalfa, and during the winter feeding they were given a CJ mixture, beets, oats, dry pulp, briquette haylage, and hay.”

Lines: 170-175

Reviewer Comment

Line 291 “However, on there is no such study for the Suffolk sheep breed.” Please revise

Answer

Re-written as;

“However, there is no such study for the Suffolk sheep breed on this aim.”

Line: 351

Reviewer Comment

The available data could be much better used, the results described in a way conducive to the conclusion the authors consider.

Answer

Necessary improvements were done.

Lines: 363-367; 371-372; 379-381; 384-387; 393-396; 404-407; 412-413; 417-429;

Reviewer Comment

The use of statistically innovative methods could be much better explained, leading, again to the conclusion. The discussion is a line of the results/conclusions from the literature and has very little merit in comparing personal results with the cited data.

Answer

Necessary improvements were done.

Lines: 363-367; 371-372; 379-381; 384-387; 393-396; 404-407; 412-413; 417-429;

Reviewer Comment

The article needs serious improvement of the English language (see the attached document), some of the sentences being very difficult to understand and misleading; in some places such sentences diminish the scientific value of the research.

Answer

Checked and corrected throughout the text.

Round 2

Reviewer 2 Report

At this second review, I would still like to reccommend the authors to finetune their English language